# Prevention and Treatment of Obesity-Related Inflammatory Diseases by Edible and Medicinal Plants and Their Active Compounds

Bashar Saad [1,2]

1    Al-Qasemi Academic College, Baqa Algharbiya 30100, Israel; bashar@aauj.edu or bashar@qsm.ac.il
2    Department of Biochemistry, Faculty of Medicine, Arab American University, Jenin P.O. Box 240, Palestine

**Abstract:** Obesity, defined by excessive fat mass and its associated low-grade chronic inflammation, leads to insulin resistance, diabetes, and metabolic dysfunctions. The immunomodulatory properties of natural agents have gained much interest in recent decades. Some of the plant-derived agents are known to be immunomodulators that can affect both innate and adaptive immunity, e.g., thymoquinone, curcumin, punicalagin, resveratrol, quercetin, and genistein. Natural immunomodulators may contribute to the treatment of a number of inflammatory diseases, as they have significant efficacy and safety profiles. The immunomodulatory effects of traditional Greco-Arab and Islamic diets and medicinal plants are well acknowledged in abundant in vitro studies as well as in animal studies and clinical trials. This review highlights the role of Greco-Arab and Islamic diets and medicinal plants in the management of inflammation associated with obesity. Although previously published review articles address the effects of medicinal plants and phytochemicals on obesity-related inflammation, there is no systematic review that emphasizes clinical trials of the clinical significance of these plants and phytochemicals. Given this limitation, the objective of this comprehensive review is to critically evaluate the potential of the most used herbs in the management of obesity-related inflammation based on clinical trials.

**Keywords:** obesity; inflammation; cardiovascular disease; metabolic disease; medicinal plants

## 1. Introduction

Obesity results from an imbalance in metabolism, which in turn results in an increased gain in adipose tissue [1–3]. A large body of research in recent decades has shown that abnormally high accumulation of body fat contributes to the observed significant increase in a variety of age-related inflammatory diseases. These include insulin resistance and associated type 2 diabetes (T2DM), hypertension, hyperinsulinemia, and dyslipidemia characterized by increased levels of free fatty acids and low-density lipoprotein (LDL) as well as reduced levels of high-density lipoprotein (HDL) levels. The progression of these conditions may eventually lead to atherosclerosis, making metabolic syndrome a significant risk for coronary heart disease [1–7]. The obesity condition is characterized by what is called "low-grade systemic inflammation" triggered by various inflammatory mediators [1–7].

Numerous scientific papers published in the last decade highlight the immunological role of adipocytes and their role in inflammatory responses through the secretion of adipocytokines (adipokines), which regulate the adipocyte phenotype through complex mechanisms of action. Normally, adipose tissue produces anti-inflammatory mediators, but with increasing cell hypertrophy, adipose tissue secretes a number of pro-inflammatory cytokines and hormones, such as tumor necrosis factor-$\alpha$ (TNF-$\alpha$), interleukin-6 (IL-6), plasminogen activator inhibitor-1 (PAI-1), angiotensinogen, transforming growth factor-$\beta$ (TGF-$\beta$), leptin, adiponectin, resistin, and monocyte chemoattractant protein-1 (MCP-1). They also produce the pro-inflammatory hormone leptin, which inhibits the secretion of the anti-inflammatory hormone adiponectin. Compared with subcutaneous adipose tissue

(SAT), visceral adipose tissue (VAT) has a higher rate of lipolysis, a higher infiltration rate of macrophages, and a higher secretion of IL -6, MCP-1, and other inflammation-related markers. With increasing obesity, monocytes infiltrate into adipose tissues, where they mature into macrophages [8–12].

In the last two decades, chronic low-grade inflammation has been repeatedly shown to cause T2DM. Inflammation decreases insulin sensitivity, leading to hyperglycemia and eventually to T2DM and its related diseases. Overweight (BMI over 25) and obese (BMI over 30) individuals generally have a higher prevalence of T2DM. Adipocytes, especially those located around the waist, secrete pro-inflammatory mediators [13–15]. Although various cellular and molecular pathways by which pro-inflammatory cytokines decrease insulin sensitivity in obese individuals have recently been investigated, the development and progression of obesity-induced insulin resistance remain to be studied in detail [4,16]. Studies currently focus on macrophages in adipose tissue as the main mediators of inflammation, but the role of metabolic changes in the liver, pancreas, and muscle also remains to be investigated [8–16].

In the last three decades, weight loss has been considered one of the central approaches to managing obesity and its associated diseases. Weight loss can be achieved through increased physical activity, reduced food consumption, and behavioral changes. These interventions have been insufficient in overweight and obese individuals over the long term. The majority of people take back their original weight within 5 years. Therefore, recent trends in obesity treatment focus on finding safe and effective natural products that promote weight loss [4,16].

Dietary and herbal products for weight loss are the most sought-after treatments in traditional medicine. A large number of crude plant extracts, isolated phytochemicals, and plant mixtures are used for weight loss. Herbal products contain a variety of phytochemicals that are effective in preventing obesity and its associated oxidative stress and low-grade chronic inflammation; people believe that a natural medicine must also be safe and effective. However, data obtained in recent years shows that this is not always the case. Many of the weight loss preparations on the market based on natural products are not well controlled, if at all. In many cases, the use of these products has resulted in serious health complications and even death [16–23].

## 2. Management of Obesity by Natural Products and Medicinal Plants

At least 25% of the active compounds of modern pharmaceutic drugs were prepared from herbs. Disappointment with the relatively high costs and possible side effects of conventional pharmaceuticals have led to the currently perceived common use of herbal-based remedies. The health-promoting properties of black seeds, cumin, pomegranate, citrus fruits, rosemary, ginger, olive, turmeric, cinnamon, and the Mediterranean diet have been confirmed in cell culture test systems, animal test models as well as in clinical investigations [4,16]. Extracts and active compounds isolated from black seed, green tea, fenugreek, and garlic were shown to exhibit direct effects on adipose tissue [23–25].

Besides the use of the intact herb or its extracts, a vast number of scientific papers support the anti-obesity properties of phytochemicals (Figure 1) [4,16,22–25]. For example, polyphenols, one of the ubiquitous groups of secondary plant products, are widely found in fruits, vegetables, cereals, legumes, and wine. Luteolin and apigenin, kaempferol, quercetin and myricetin, diadzein and genistein, cyanidin, grape seed proanthocyanidin extract, xanthohumol and epigallocatechin gallate (EGCG) are examples of polyphenols, and their weight-reducing properties, as well as their anti-obesity effects, are well documented in a large number of scientific papers [4,16,22–25].

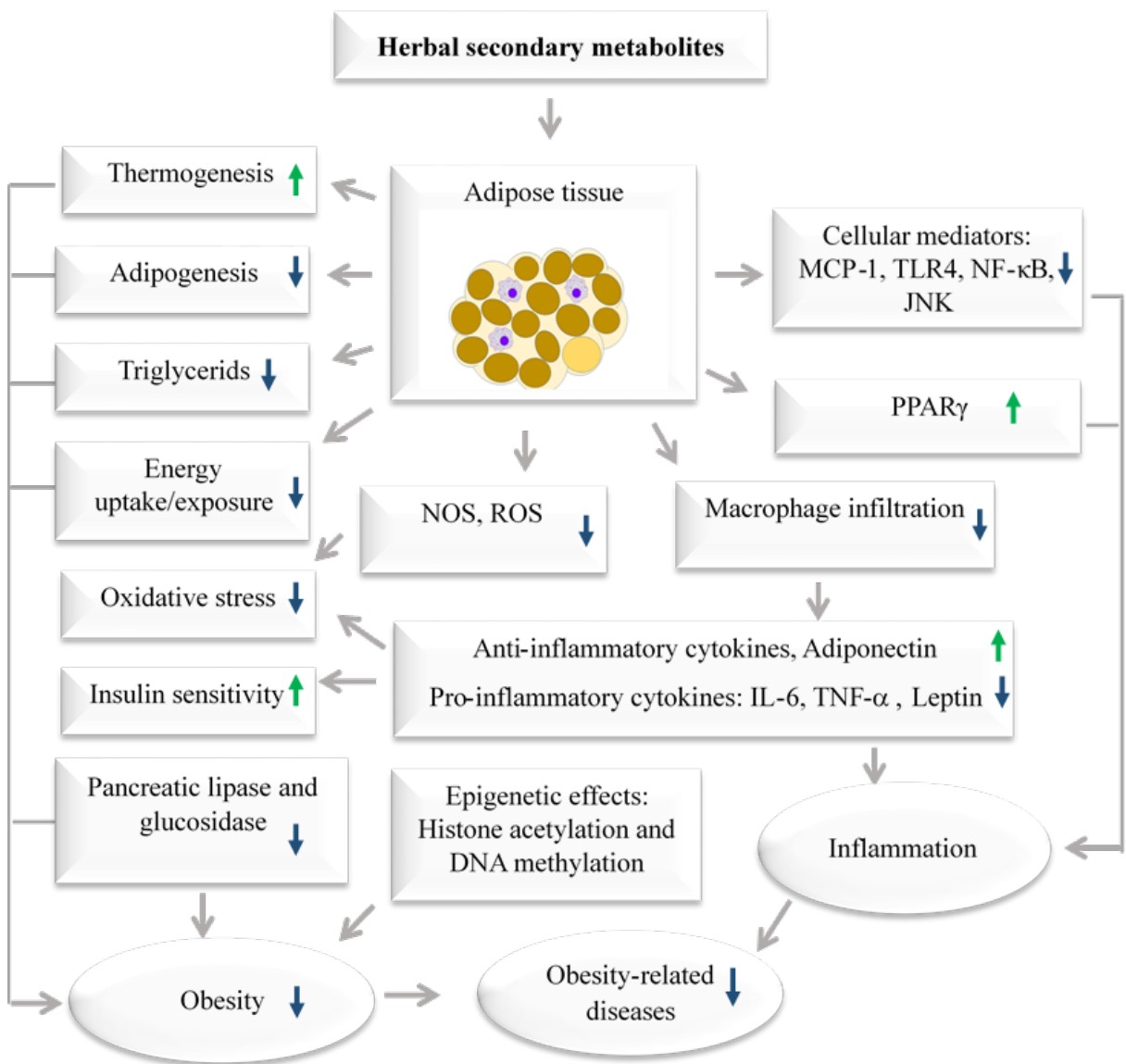

**Figure 1.** Management of overweight and obesity by medicinal plants, diet, and natural products.

Restraining appetite, reducing pancreatic lipase activity, repressing adipogenesis, activation of lipolysis, thermogenesis, and lipid metabolism are the main targets for weight-reducing management (Figure 1).

*Low-Grade Chronic Inflammation in Adipose Tissue*

The pathogenesis of obesity-related chronic diseases is caused by the low-grade chronic inflammation in adipose tissue and systemically. At the cellular level, brown adipose tissue (BAT) and white adipose tissue (WAT) are composed of pre-adipocytes, mature adipocytes, fibroblasts, and infiltrated macrophages, among others. Adipocytes produce leptin and adiponectin, as well as chemokines that influence cell behavior, inflammation, and the tumor microenvironment. In general, adipose tissue primarily secretes anti-inflammatory molecules, but with increased fat mass, cells in adipose tissue begin to secrete pro-inflammatory cytokines (TNF-α, IL-1β, IL-6, IL-10, CRP, iNOS, and MCP-1) and leptin. Increased fat metabolism leads to increased levels of free fatty acids, which initiate inflammatory signaling cascades in infiltrating macrophages, dendritic cells, and T lymphocytes. A feedback loop of pro-inflammatory mediators reinforces this inflamma-

tory state and further drives macrophage migration and cytokine production, leading to disruption of the insulin signaling cascade and insulin resistance [9,26–32].

Both adipocytes and infiltrated macrophages are the main sources of pro-inflammatory mediators [27–32]. Therefore, targeting adipose tissue is a valuable tool for the treatment of obesity-related diseases. Numerous publications have reported that medicinal plants and phytochemicals can reduce inflammatory responses in adipose tissue through several pathways. These include inhibiting the production of pro-inflammatory cytokines, increasing the production of anti-inflammatory cytokines, peroxisome proliferator-activated receptor $\gamma$ (PPAR$\gamma$), and reducing c-Jun amino-terminal kinase (JNK) and/or nuclear factor-$\kappa$B (NF-$\kappa$B) signaling [27,32]. The latter represents a potential target for the development of new anti-inflammatory drugs. It plays an important role in signaling pathways regulating the production of pro-inflammatory cytokines, chemokines, and adhesion molecules.

Toll-like receptors (TLRs) bind to pathogen-associated molecular patterns (PAMPs/DAMPs) and initiate innate immune responses [26–32]. Binding of TLR4 to the lipopolysaccharide (LPS)-LPS-binding protein (LBP)-CD14 complex initiates an intracellular signaling pathway that involves the activation of p38, JNK, and ERK MAPKs and regulates gene expression through NF-$\kappa$B activation. Both NF-$\kappa$B and JNK play an important role pro-inflammatory gene activation downstream of TLRs in adipose tissue, so targeting the TLR4/NF-$\kappa$B or TLR4/JNK axis may be useful to inhibit inflammatory responses in adipose tissue and associated insulin resistance in obese individuals [26–32].

Oxidative stress occurs when the levels of reactive oxygen species (ROS) and reactive nitrogen species (NOS) exceed antioxidant capabilities. ROS and NOS play an important role upstream and downstream of NF-$\kappa$B and TNF-$\alpha$ signaling. Nuclear factor erythroid-2 related factor 2 (Nrf2) is strongly associated with inflammation-associated oxidative stress. Its role has been elucidated in the kidney and heart in a model of chronic kidney injury and in models of quinolinic acid-induced neuronal injury and in vitro with granule neurons. There are also investigations showing mutual regulation of Nrf2 and NF-$\kappa$B, suggesting its anti-inflammatory role. In addition, numerous papers have shown that it is linked with MAPK, NF-$\kappa$B, PI3K, and PKC signaling [33–35].

PPAR$\gamma$, a member of the nuclear receptor superfamily, control lipid and carbohydrate metabolism and contribute to the inflammatory response. Thiazolidinediones (TZDs), synthetic ligands for PPAR$\gamma$, are anti-diabetic agents. Activation of PPAR$\gamma$ by TZDs promotes adipocyte differentiation. PPAR$\gamma$ stimulates glucose uptake in mature adipocytes, induces secretion of adiponectin and inhibits TNF-$\alpha$ production through the PPAR$\gamma$ activation in adipocytes. Therefore, natural PPAR$\gamma$ ligands represent a valuable tool in the treatment of inflammation and T2DM [32–37]. Many herbs and phytochemicals have been shown to attenuate inflammatory responses in adipose tissue. Therefore, the search for plant secondary metabolites as new targets for the development of effective and safe anti-inflammatory agents is attracting much research interest [38–45]. In the following section, we will present the most commonly investigated anti-obesity medicinal plants that downregulate obesity-related low-grade inflammation.

## 3. Medicinal Plants with Immunomodulatory Properties

### 3.1. Curcuma Longa Rhizomes (Turmeric)

Curcuma longa rhizomes (Turmeric) have a long history of use as a health-promoting agent [21,22,24,46–49]. The roots of turmeric have been known for their health values for hundreds of years in Ayurveda, Chinese, and Greco-Arab and Islamic medicine [50,51]. Over 800 scientific reports and more than 100 clinical trials have investigated the cellular, molecular, and pharmacological effects of curcumin. Many of these reports emphasize the potential benefits of curcumin in the treatment of chronic diseases such as cardiovascular disease, T2D, overweight/obesity, cancer, autoimmune diseases, neurological and mental disorders [39–42,46]. Most published reports have linked the benefits of turmeric to its antioxidant and anti-inflammatory properties [52].

Curcumin, the major polyphenol from turmeric, has been reported to promote weight loss and reduce the incidence of obesity-related diseases [41–43]. Curcumin intake reduces levels of TNF-$\alpha$ and IL-6 and increases the level of adiponectin in the plasma of obese and overweight individuals. In addition, curcumin regulates a number of biochemical and molecular targets, including transcription factors (NF-kB, NLP3), signaling pathways, and other complex regulatory systems, resulting in attenuation of the chronic low-grade inflammatory response in adipose tissue.

Several clinical studies have shown that the interaction of curcumin with transcription factors, cellular receptors, growth factors, enzymes, cytokines, and chemokines reduces inflammation in obese individuals by restoring the balance between pro-inflammatory and anti-inflammatory mediators [53,54]. Several published studies clearly indicate that curcumin significantly decreases pro-inflammatory cytokine levels and increases plasma adiponectin levels in obese and overweight individuals. Furthermore, curcumin can affect multiple molecular targets, including transcription factors (NF-kB, NLP3), signaling pathways, and other complex regulatory systems in adipose tissue, resulting in the attenuation of chronic low-grade inflammatory response. In addition, there are some reports suggesting that curcumin enhances the effect of diet and lifestyle intervention in overweight/obese people with metabolic syndrome [55,56]. However, the poor bioavailability of curcumin poses a problem for its use. To increase the bioavailability of curcumin, various delivery systems, such as micelles, liposomes, phospholipid complexes, nanostructured lipid carriers, and biopolymer nanoparticles have been developed [38,41–43,57–59]. The future of curcumin as an approved option for the prevention or treatment of obesity and its associated low-grade inflammation and other complications depends on the results of high-quality and large cohort studies in the future. However, based on current knowledge, curcumin has a good safety profile and is well tolerated. This low toxicity may be due to the low bioavailability, therefore, an increase of biodisponibility must be carried out cautiously.

### 3.2. Camellia sinensis (Tea Leaves)

*Camellia sinensis* (Tea leaves) are the source of white, green, and black tea. Several observational clinical trials have shown that the consumption of tea or green tea has a beneficial effect against obesity. The effects observed in these studies are low, so statistical confirmation is especially difficult in rather healthy people. For example, a cross-sectional study of 1210 adults showed that regular tea drinkers for more than 10 years showed a 19.6% reduction in body fat percentage and a 2.1% reduction in waist-to-hip ratios when compared with irregular tea drinkers [60]. Recent results from a clinical trial of 6472 adults showed that tea consumers had a lower average waist circumference and body mass index (BMI) than non-consumers [61]. However, a more recent study of 3539 participants showed that green tea was not associated with visceral obesity or metabolic syndrome [62]. Therefore, further studies are needed to clarify the anti-obesity properties of green tea.

Epigallocatechin-3-gallate (EGCG), derived from green tea, and theaflavins, derived from black tea, are the best-studied active compounds from tea. EGCG, the main polyphenolic compound in green tea, has a number of health-promoting effects. These include anti-inflammatory, antioxidant, anti-obesity, anti-cancer, and anti-diabetic properties. The anti-inflammatory effects of EGCG, such as attenuating the secretion of resistin from adipocytes, are exerted through the ERK-dependent pathway. EGCG-enhanced production of adiponectin is mediated, at least in part, by inhibition of the protein Krueppel-like factor 7 (KLF7), which downregulates the expression of genes controlling adipogenesis [46,63–71]. EGCG also reduced inflammation and oxidative stress associated with aging in high-fat diet-induced rats. EGCG significantly reduced systemic levels of IL-6, tumor necrosis factor-$\alpha$ (TNF)-$\alpha$, ROS, and superoxide dismutase (SOD). EGCG also increased the expression of sirtuin-1 (SIRT1), catalase, fatty acid-binding protein-1, glutathione S-transferase (GST)-A2, and acyl-CoA synthetase-1, but significantly decreased the expression of nuclear factor (NF)-$\kappa$B, ACC -1, and FASN in the liver [72].

### 3.3. Capsaicin

A fat-soluble alkaloid, extracted from Capsicum (hot pepper) fruits is the major bioactive compound in red chili peppers. It is a pungent molecule that affects thermoregulation, triggers autonomic reflexes, and is well absorbed [73]. Capsaicin is in the pipeline for phase III clinical trials as a treatment option for rheumatoid arthritis, postoperative pain, and chronic neuropathic and musculoskeletal pain [73]. In recent decades, the pharmacological benefits of capsaicin and its underlying mechanisms have been extensively studied. The main beneficial medicinal properties of capsaicin include analgesic, antioxidant, anti-inflammatory, anti-cancer, anti-obesity, cardio-protective, and metabolic modulatory effects [74–77]. Capsaicin binds to Transient Receptor Potential Channel Vanilloid type-1 (TRPV1) [10,11], a transmembrane ion channel [78]. TRPV1 is expressed in many cell types and tissues, including nerve fibers, trigeminal ganglia, testis, adipocytes, smooth muscle cells, endothelial cells, pancreatic β-cells, liver, heart, skeletal muscle, and kidneys [79]. Selective silencing of TRPV1 by specific RNA interference reduced the effect of capsaicin on calcium influx and inhibition of adipogenesis in 3T3-L1 adipocytes [80].

Several metabolic studies have shown that capsaicin is a potent anti-inflammatory substance [81–83] that could attenuate metabolic inflammatory conditions such as obesity, T2DM, osteoarthritis, and non-alcoholic fatty liver disease. Capsaicin has been shown to attenuate (via blocking NF-κB, which is likely mediated by PPARγ activation) the expression and secretion of MCP-1 and IL -6 from adipose tissue of mice and to increase the production of adiponectin. It has also been shown that capsaicin administration in vivo improves obesity-induced insulin resistance [46,74–77]. Moreover, in healthy rats, capsaicin decreased oxidative stress measured by malondialdehyde and diene conjugation in tissues [84]. Capsaicin prevented lipid peroxidation and carbonyl formation in proteins in human erythrocytes subjected to oxidative stress [85].

### 3.4. Zingiber Officinale (Ginger)

An herbaceous perennial plant of the family Zingiberaceae, it is one of the most famous medicinal herbs in traditional Greco-Arab and Islamic medicine, Ayurveda, and Chinese medicine for centuries. It is reported to have several health-promoting properties. These include antiulcer, anti-inflammatory, antioxidant, antiplatelet, anti-diabetes, anti-obesity, anti-hyperlipidemia, as well as cardiovascular and anti-cancer activities. The anti-obesity activities of ginger are most likely due to the highly significant inhibitory action of ginger on the absorption of dietary fats by reducing pancreatic lipase activity. Ginger contains active compounds such as gingerol, paradol, and 6-shogaol responsible for the anti-inflammatory effects of ginger. 6-shogaol elevates adiponectin production and inhibits the TNF-α-induced downregulation of adiponectin production in adipocytes, most likely by upregulating PPARγ activity. As for 6-shogaol, 6-gingerol inhibits TNF-α-mediated inhibition of adiponectin in adipocytes; however, the pathways of their inhibitory effects are different; 6-gingerol inhibits JNK signaling pathways in TNF-α-induced adipocytes without affecting PPARγ transactivation, whereas the anti-inflammatory effects of 6-shogaol is PPARγ-dependent [4,16,27,86].

Systematic review and meta-analysis of 16 randomized controlled trials RCTs comprising 1010 participants provide convincing evidence for a significant effect of ginger in lowering circulating C-reactive protein (CRP), high sensitivity C-reactive protein (hs-CRP), and TNF-α levels. Large-scale RCTs are still needed to draw concrete conclusions about the effect of ginger on other inflammatory mediators [87].

### 3.5. Nigella sativa (Black Seeds)

*Nigella sativa* (Black seeds), also called black cumin or black seed, is famous for its culinary uses and was historically valuable in traditional Greco-Arab and Islamic medicine and other traditional medicines. Prophet Muhammad (PBUH) (570–632 AD) said, "Black seed can cure any disease except death." Avicenna (980–1037 AD) mentioned black seed in his canon of medicine as "the seed that stimulates the energy of the body and promotes

recovery from fatigue and dejection" [4,16,88–90]. Numerous scientific papers have been published addressing the safety and efficacy of black seed and thymoquinone (the main active ingredient). A "Medline" and "Google Scholar" search with "black seed"/"*Nigella sativa*" or "thymoquinone" yields more than 1100 publications.

*Nigella sativa* exhibits pleiotropic pharmacological activity and a wide range of health-promoting effects (Figure 2). Both the seeds and their major bioactive constituent thymoquinone have been found to exert significant antioxidant, anti-inflammatory, immune enhancing, cell survival improving, and energy metabolism promoting effects underlying various health benefits. These include protection against metabolic, cardiovascular, digestive, hepatic, renal, respiratory, reproductive, and neurological diseases, T2DM, obesity, hypotension, allergies, antimicrobial effects, and cancer [24,25]. Despite significant advances in pharmacological benefits, these black seeds and their active compounds are still far from the clinical application [91].

Black seed and thymoquinone attenuate the obesity-related low-grade inflammation through activation of natural killer cells proliferation and differentiation, monocyte activities, T-cell based immunity, as well as stimulation macrophage activity. For example, black seed significantly attenuates nitric oxide production and serum levels of pro-inflammatory cytokines and mediators, including IL-4, IL-5, IL-6, IgE, IgG1, and OVA-specific IgG1 ovalbumin-treated rats. Rats treated with black seed experienced a reduced T-cell response and attenuated T-cell proliferation in the spleen without histopathological changes in lung tissue. Control, untreated rats exhibited a thickening of the alveolar wall and increased numbers of goblet cells. These data suggest that black seed inhibits Th-2-induced T-cell proliferation and differentiation, thereby halting the inflammatory response [92–94]. Pre-treatment with thymoquinone attenuated Th-2-cell mediated lung inflammation, lung eosinophilia, and goblet cell hyperplasia. Furthermore, thymoquinone downregulated COX-2 expression, PGD2 production, and a slight inhibition in COX-1 expression and PGE2 production in rats. COX-2 mediates the inducible inflammatory response by converting arachidonic acid to prostaglandins (a pro-inflammatory cytokine), whereas COX-1 mediates constitutive or "housekeeping" inflammation. Long-term elevated COX-2 activity is recognized as an underlying cause of many chronic inflammatory disorders, thus, inhibition of COX-2 is favorable in cases of chronic inflammatory conditions. Black seeds and thymoquinone exert their anti-inflammatory effects primarily via downregulation of COX-2 and PGD2 production [88–90].

*Nigella sativa* oil reduced the levels of IL-6 and IL-1$\beta$ in human pre-adipocytes [92]. Treatment with *Nigella sativa* oil (400 mg/kg) in rats with carrageenan-induced paw edema, improved the pro-inflammatory cytokines IL-6, IL-12, and TNF-$\alpha$ in paw exudates and sera [93,94]. Furthermore, topical application of balm stick containing 10% *Nigella sativa* oil in rats with paw edema markedly alleviated acute and sub-acute inflammation with a significant edema reduction, with a 43% lower leucocytes count and 50% lower TNF-$\alpha$ level compared to control on the inflammation area [95].

Overall, the evidence supports the anti-inflammatory potentials of black seeds and thymoquinone, however, most of the studies so far have been conducted in animal models. Future clinical trials should focus on determining the anti-inflammatory potential in ameliorating human diseases

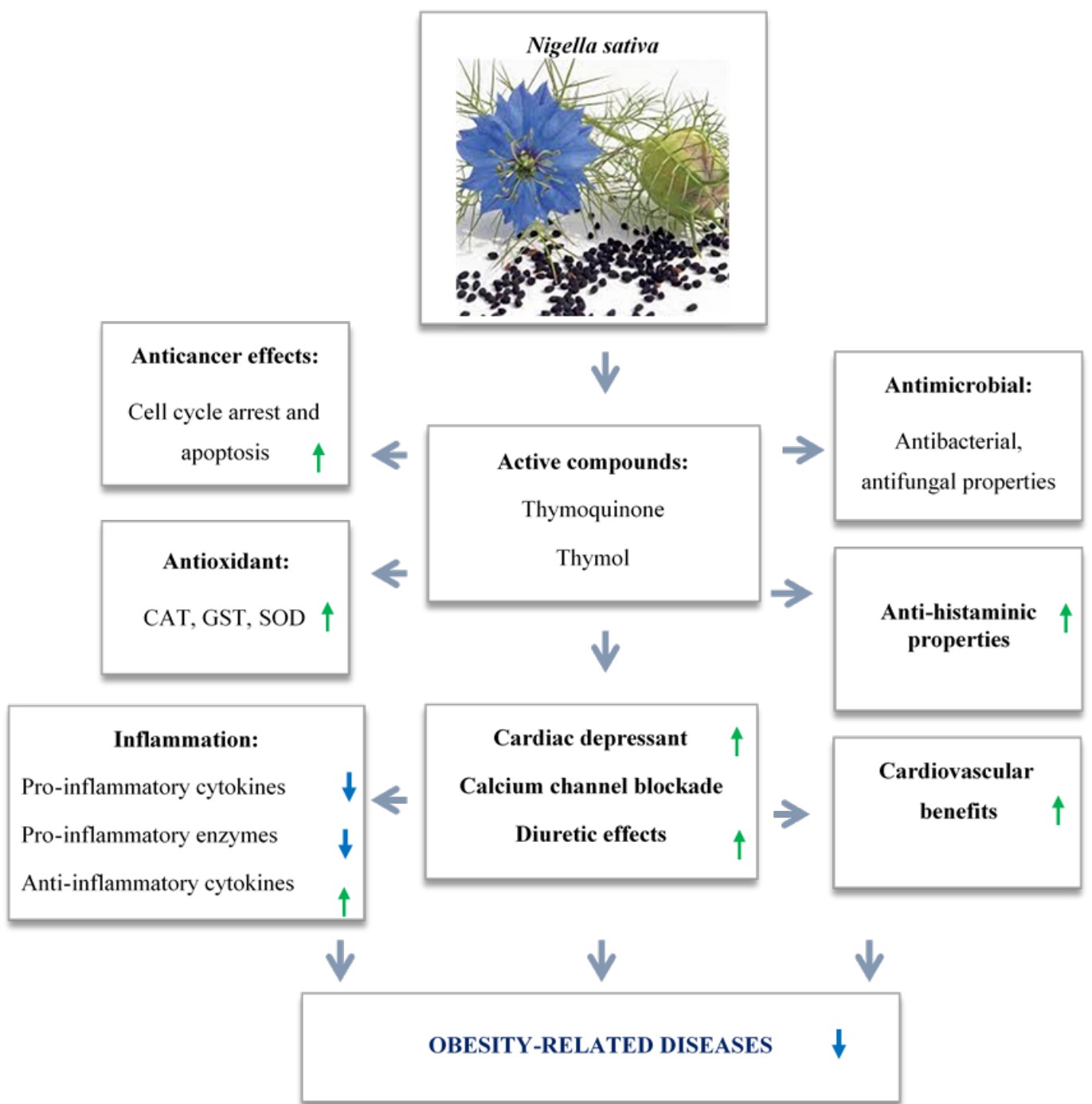

**Figure 2.** Management of obesity-related diseases by black seeds and their derived active compounds. SOD: superoxide dismutase, CAT: catalase, GST: glutathione S-transferase.

### 3.6. Punica granatum (Pomegranate)

*Punica granatum* (pomegranate) has long been used in major traditional medicine systems to promote health and treat many diseases. In recent decades, a large number of scientific papers have been published on the health-promoting effects of pomegranate. These include antioxidant, anti-inflammatory, cardiovascular, anti-obesity, anti-diabetic, and anti-cancer effects (Figure 3).

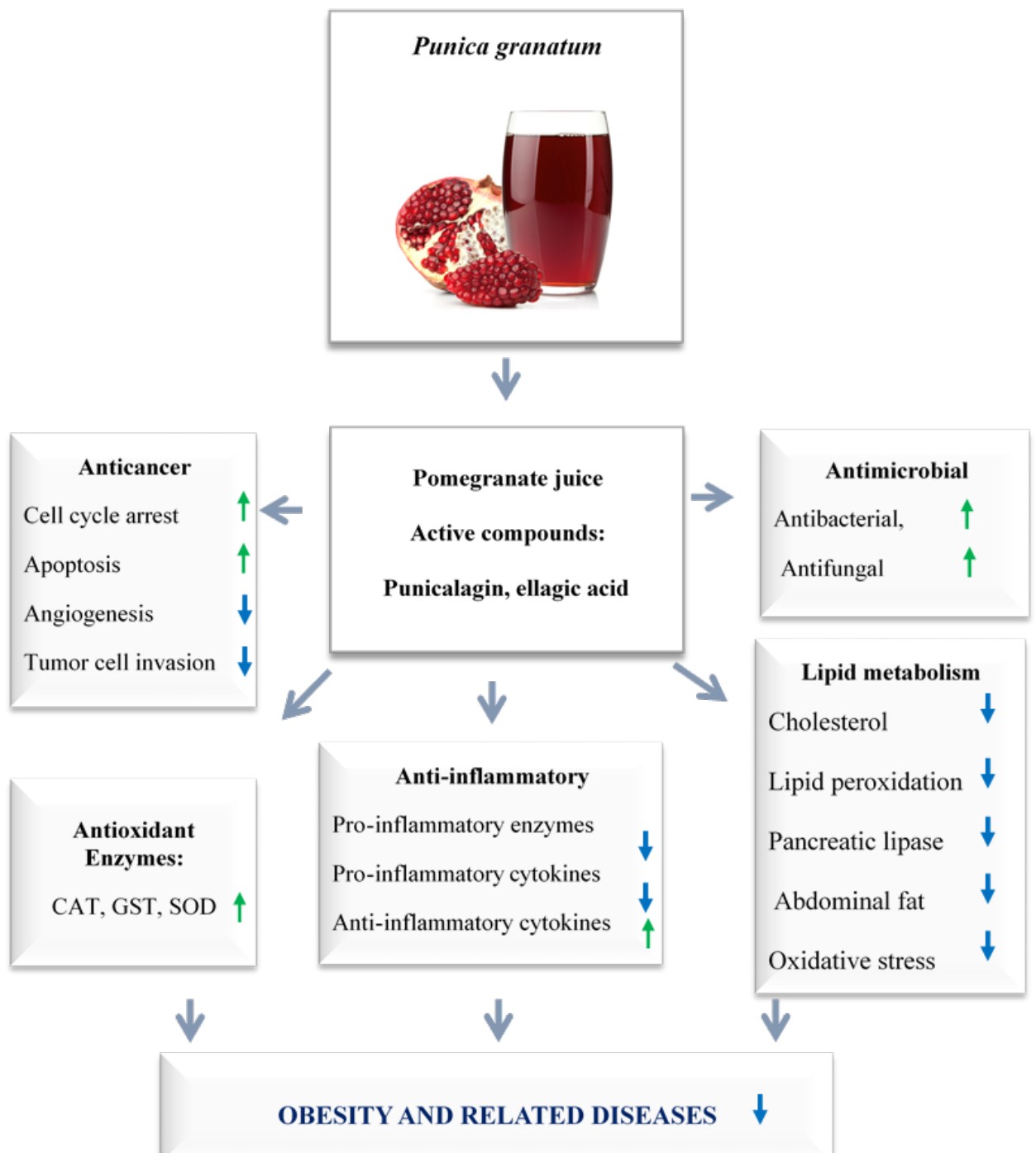

**Figure 3.** Management of obesity-related diseases by *Punica granatum* and its derived active compounds. SOD: superoxide dismutase, CAT: catalase, GST: glutathione S-transferase.

Pomegranate fruits and other aerial parts of the plant contain several bioactive molecules. These include phenolic acids, hydrolyzable tannins, condensed tannins, and flavonoids, as well as other types of bioactive constituents responsible for the observed antimicrobial, anti-cancer, antioxidant, and anti-inflammatory activities. Pomegranate-derived punicalagin isomer, ellagic acid, and anthocyanins are known for their antioxidant properties and reduction of lipid oxidation. Pomegranate extracts and juice are also effective in stimulating vascular endothelial NO synthase and plasma NO levels, suggesting clinical application in metabolic syndrome [4,16,24,25,96].

As mentioned earlier, obesity is associated with elevated levels of adipocyte-derived pro-inflammatory cytokines such as TNF-a and IL-6, which affect metabolism in several ways. Pomegranate affects adipocyte-specific gene expression, triacylglycerol release, lipoprotein lipase downregulation, and insulin sensitivity [96–99]. Pomegranate active compounds have been shown to reduce the secretion of IL-6, thereby reducing the complications associated with obesity. Thus, in vitro experiments reported that pomegranate seed oil (PSO) decreased the activities of cyclooxygenase and lipoxygenase. The activity of cyclooxygenase, a key enzyme in the conversion of arachidonic acid to prostaglandins (important mediators of inflammation), was reduced to 60% by PSO. Lipoxygenase, which catalyzes the conversion of arachidonic acid to leukotrienes, also important mediators of inflammation, was inhibited by PSO by 75% compared to controls [96]. The effects of pomegranate extract consumption on plasma inflammation, oxidative stress biomarkers, and serum metabolic profiles were investigated in a randomized, double-blind, placebo-controlled clinical trial in overweight and obese subjects. In this study, 48 overweight and obese participants were randomized to receive either 1 g of pomegranate extract or placebo, daily for 30 days. Ingestion of pomegranate extract resulted in significant reductions in mean serum levels of glucose, insulin, total cholesterol, LDL-C, plasma malondialdehyde (biomarker of oxidative stress), and IL-6. These results suggest that the consumption of pomegranate extract may reduce the complications associated with obesity [95–99].

A systematic review of 20 clinical trials suggests that pomegranate and its active ingredients lower BMI, hypertension, blood glucose levels, triglycerides, total cholesterol, and LDL. It may also increase HDL levels and improve insulin resistance. Although relevant effects have been observed, further well-designed clinical studies are needed to determine the proper formulations and dosages that can be used to prevent or treat metabolic syndrome components [100].

Regarding the anti-inflammatory effects of pomegranate, a recent review article considered 16 randomized controlled trials (RCTs) involving 572 subjects [101]. Pomegranate intake significantly lowered levels of hs-CRP, IL-6, and TNF-α compared with placebo. No significant reduction was observed in CRP, E-selectin, ICAM, VCAM, or MDA compared to placebo [101].

### 3.7. Grapes, Peanuts, and Many Berries

Grapes, peanuts, and many berries contain resveratrol, a nonflavonoid that belongs to the stilbene group. Resveratrol is a promising phytochemical that can be easily incorporated into the diet to combat adipose tissue inflammation and other obesity-related metabolic diseases. There are several lines of evidence that resveratrol has antiadipogenic effects (Figure 4). In vitro studies show that resveratrol stimulates apoptosis in mature adipocytes and targets triacylglycerol metabolism at WAT. These effects appear to be mediated by the attenuation of fatty acid uptake and lipogenesis in adipose tissue. In addition, the increase in BAT thermogenesis and associated energy depletion may help explain the body fat-lowering effects of resveratrol. However, a meta-analysis of 19 clinical trials found that only three studies showed any type of beneficial effect. The meta-analysis found no significant effect on weight or BMI. A small effect was found on waist circumference [102].

Resveratrol is a potent anti-inflammatory phytochemical that attenuates the activity of NF-κB and ERK. In mouse adipose tissue, resveratrol attenuated the production of TNF-α, interferon α, and (IFNα and IFNβ), and IL -6 and their upstream signaling molecules, including TLR 2/4 triggered by a high-fat diet, Toll IL-1 receptor (TIR) domain-containing adaptor protein (TIRAP), TIR domain-containing adapter-inducing interferon (TRIF), TNF receptor-associated factor 6 (TRAF6), interferon regulatory factor 5 (IRF5), p-IRF3, and NF-κB. In addition, improvement of insulin sensitivity and attenuation of inflammation by resveratrol are mediated via inhibition of adipokine production (e.g., resistin and retinol-binding protein 4), attenuation of oxidative stress, and stimulation of Akt-mediated insulin signaling [26,27,46,103,104].

A systematic review of 17 RCTs involving 736 subjects found significant reductions in TNF-α and hs-CRP levels after resveratrol supplementation. Resveratrol supplementation had no significant effect on the levels of IL-6. Statistically, significant heterogeneity was observed in relation to the type of sample IL-6 and study duration for the inflammatory markers IL-6, TNF-α, and hs-CRP [105].

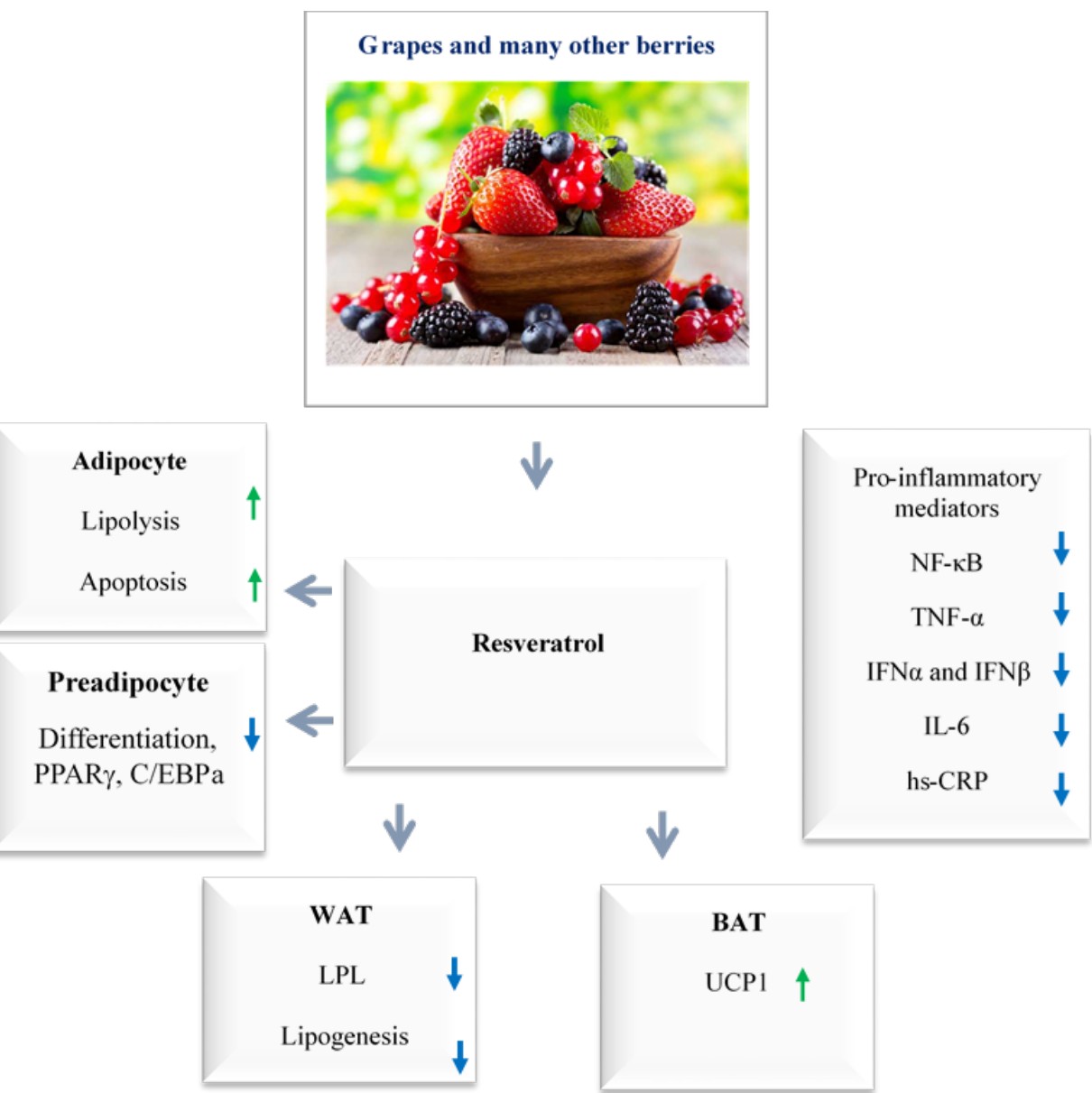

**Figure 4.** Action mechanisms of anti-obesity effects of resveratrol. BAT: brown adipose tissue; C/EBP: CCAAT/enhancer-binding protein; LPL: lipoprotein lipase; PPAR: peroxisome proliferator-activated receptor; UCP: uncoupling protein; WAT: white adipose tissue. NF-κB: nuclear factor-κB; TNF-α: tumor necrosis factor-α; IL-6: interleukin-6; IFN: interferon; hs-CRP: high sensitivity C-reactive protein.

## 4. Anti-inflammatory and Antioxidant Potential of Polyphenols Contained in Mediterranean Diet in Obesity

The inhabitants of the 16 countries bordering the Mediterranean belong to different nations and religions. Although their diet varies according to cultural and religious beliefs, overall, they follow Mediterranean dietary patterns [106,107]. The Mediterranean diet, one of the best-studied and best-known dietary patterns in the world, is associated with

a variety of health-promoting effects. This diet is characterized by a high proportion of fruits, vegetables and salads, breads, and whole grains, potatoes, legumes/beans, nuts, and seeds (Figure 5). A common and central feature of all variants of the Mediterranean diet is the consumption of olive oil (monounsaturated fatty acid) as one of the main sources of fat [106,107].

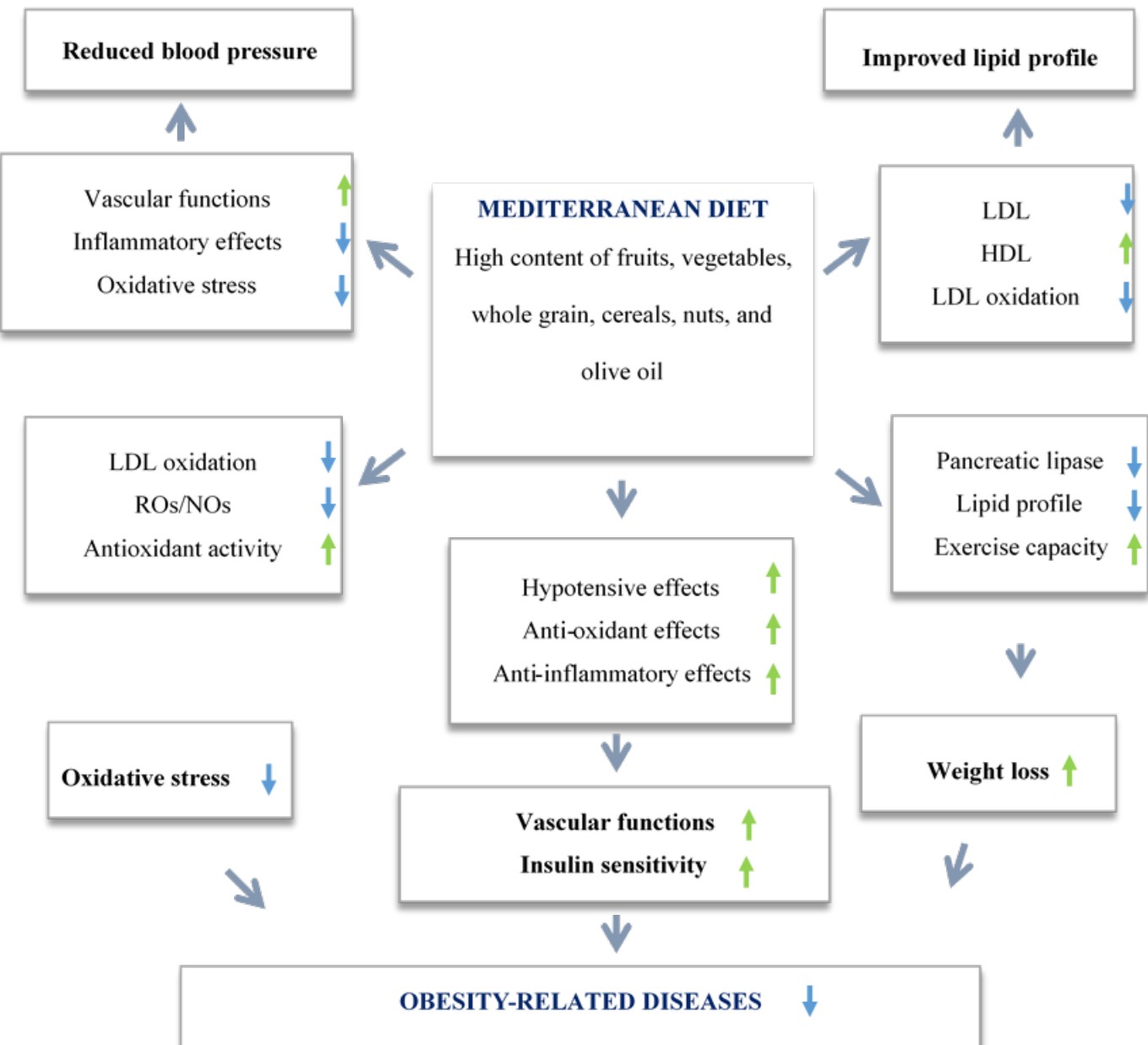

**Figure 5.** Cardiovascular beneficial effects of the traditional Mediterranean diet.

The Mediterranean diet is one of the best-studied diets for promoting cardiovascular health. It has been found to reduce or even prevent the burden of cardiovascular disease, hypertension, T2DM, obesity, breast cancer, depression, colorectal cancer, asthma, erectile dysfunction, and cognitive decline [22,24]. Several observational studies and randomized controlled trials show that this diet reduces waist-to-hip ratio, lipids, and pro-inflammation mediators [16,108]. However, it is unclear whether the cardiovascular disease benefits of this diet are due to the individual components or to the synergistic effects of the various components. These exhibit antioxidant, anti-inflammatory, and antihypertensive effects [22,24,109,110].

A recent systematic review examined the effects of the Mediterranean diet on changes in HDL structure and functionality in humans in 13 studies. The Mediterranean diet showed significant beneficial effects on HDL functionality, particularly by improving HDL cholesterol efflux capacity and reducing HDL oxidation. Thus, the Mediterranean diet is a protective factor against cardiovascular disease and has been associated with increasing HDL quality and inhibiting HDL dysfunctionality, among other benefits [111].

Olive oil is the main source of fat in the traditional Mediterranean diet. Like *Nigella sativa*, olive is one of the most widely used medicinal plants in the Mediterranean. While olive oil is known for its health benefits, olive leaf has been used medicinally in various historical contexts and cultures. Olive leaves and olive leaf extracts are now marketed as antioxidants, anti-aging agents, and immune stimulants. Clinical studies have demonstrated their anti-diabetic, antihypertensive, antibacterial, antifungal, and anti-inflammatory effects. Olive oil plays a central role in Greco-Arab and Islamic medicine. The olive tree is described in the Quran as a sacred tree and the Prophet (pbuh) said, "Eat olive oil and massage it on your body, for it is a sacred tree." [4,24].

The health-promoting effects are attributed to the consumption of olive oil, which is considered an important bioactive food due to its high nutritional quality [112]. There are a large number of reports that provide information on the tools underlying the prevention of cardiovascular disease by olive oil (Figure 6) [113,114]. These data show that the consumption of extra virgin olive oil is associated with a beneficial effect on cardiovascular disease. Intervention studies are consistent with these beneficial effects, which are supported by the ability of extra virgin olive oil to prevent or reduce inflammatory processes associated with chronic degenerative diseases such as cardiovascular disease and cancer. Consumption of a Mediterranean diet supplemented with extra virgin olive oil was also associated with significant reductions in systemic inflammatory markers (IL-6, IL-7, IL-aa18, and hs-CRP) in patients with metabolic syndrome [114,115]. Short- and long-term studies showed that supplementation with extra virgin olive oil was associated with a significant decrease in thromboxane B2 (TXB2) and leukotriene B4, confirming the antithrombotic and anti-inflammatory properties of extra virgin olive oil in the postprandial state [116]. Studies in patients at high cardiovascular risk have shown that supplementation with extra virgin olive oil lowers both systolic and diastolic blood pressure [117,118].

A systematic review and meta-analysis of RCTs was recently published on the effects of the Mediterranean diet on blood pressure [119]. This review included 19 RCTs with data from 4137 participants and 16 observational studies with data from 59,001 participants. Adherence to the Mediterranean diet reduced systolic blood pressure and diastolic blood pressure by an average of −1.4 mmHg and −1.5 mmHg, respectively (95% CI: −2.74 to −0.32 mmHg), compared with the control group. Meta-regression showed that longer study duration and higher baseline systolic blood pressure were associated with greater reductions in blood pressure in response to a Mediterranean diet. In observational studies, hypertension was 13% less likely to occur when a Mediterranean diet was followed than when it was not [119]. These data suggest that the Mediterranean diet is an effective dietary strategy to support blood pressure control, possibly contributing to the lower risk of the cardiovascular disease reported with this diet [119].

Numerous scientific publications have demonstrated the long-term overall health benefits of the Mediterranean diet. However, its effectiveness in reducing body weight over the long term (longer than one year) in overweight or obese individuals remains controversial. A systematic review of five RCTs [109] examined the effectiveness of the Mediterranean diet in reducing weight in overweight or obese individuals, comparing the Mediterranean diet with low-carbohydrate and low-fat diets and the American Diabetes Association diet. From this systematic review, the Mediterranean diet showed greater weight loss than the low-fat diets but similar weight loss to the low-carbohydrate and the American Diabetes Association diets. Similar results were also obtained in a previous meta-analysis of 16 RCTs [108], in which greater adherence to the Mediterranean diet resulted in greater weight loss compared with a control diet [109]. Several components of

the Mediterranean diet may have a positive effect on weight loss. However, the effect of the Mediterranean diet on body weight was greater when combined with an energy-reduced diet plan or increased physical activity [109].

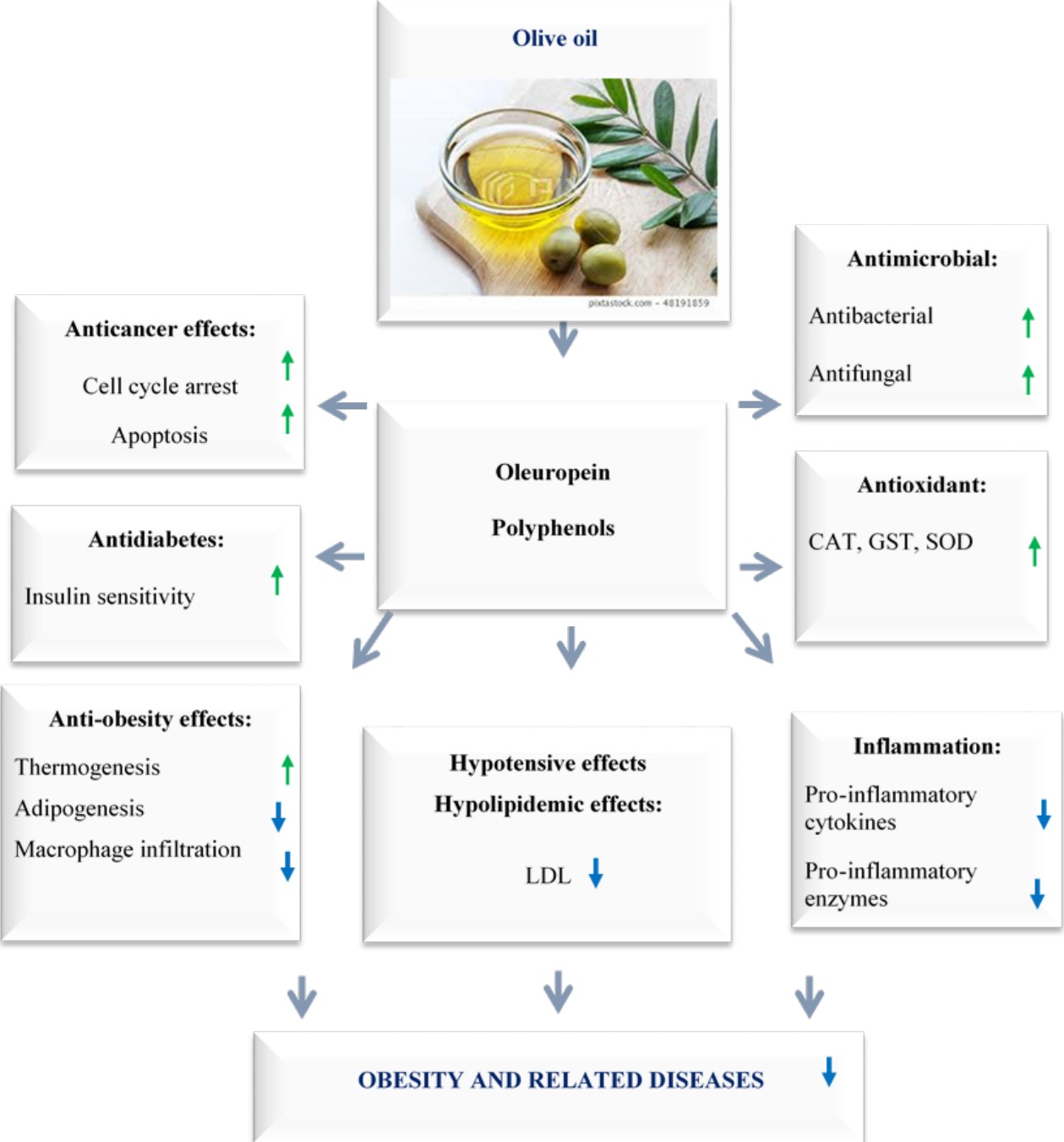

**Figure 6.** Management of obesity-related diseases by olive oil and its derived active compounds. SOD: superoxide dismutase, CAT: catalase, GST: glutathione S-transferase.

Dietary polyphenols may activate weight loss, at least in part, through stimulating β-oxidation; a prebiotic effect for gut microbiota; reducing appetite; enhancing thermogenesis in brown adipose tissue; reducing adipogenesis; increasing adipocyte apoptosis, and promoting lipolysis. Even though the intake of some specific polyphenols has been linked to weight loss, there is yet no scientific proof for the effects of total polyphenols on adiposity in clinical trials [120].

A growing body of evidence suggests that the protective effects of the Mediterranean diet are due, at least in part, to its anti-inflammatory properties [121–123]. The anti-inflammatory effects of individual components of the Mediterranean diet have been demonstrated in a large number of in vitro studies [124–127]. As mentioned earlier, low-grade chronic inflammation is associated with the development of many cardiovascular diseases and T2DM, as well as many other diseases. Therefore, studying the relationship between diet and inflammatory markers is a promising tool for the development of new anti-inflammatory drugs. A clinical trial conducted in the Balearic Islands, a Mediterranean region, investigated the association between inflammatory markers and adherence to the Mediterranean diet in adults and adolescents. This study also aimed to investigate the effect of adherence to the Mediterranean diet on inflammatory markers [127]. In this study, Sureda's group [127] examined inflammatory markers in adults and adolescents in relation to adherence to the Mediterranean diet. It included a random sample of 219 men and 379 women aged 12 to 65 years. Dietary habits were assessed and adherence to the Mediterranean diet pattern was calculated. Adherence to the Mediterranean diet pattern was 51.3% in male adolescents and 45.7% in adults, and 53.1% and 44.3% in female adolescents, respectively. In men, higher adherence to the Mediterranean diet was associated with increased adiponectin levels and lower leptin, TNF-$\alpha$, PAI-1, and hs-CRP levels in adults but not in young subjects. In women, higher adherence was associated with lower leptin levels in the young group, PAI-1 in adults, and hs-CRP in both groups. Low adherence to the Mediterranean dietary pattern is directly related to increased systemic pro-inflammatory mediators [127].

## 5. Concluding Remarks

Weight loss has been considered one of the central approaches in the prevention and treatment of obesity and its associated diseases. It can be achieved through increased physical activity, reduced food intake, and behavioral changes. However, these interventions have been insufficient in the long term. The majority of people take back their original weight within 5 years. Therefore, recent trends in obesity treatment focus on finding safe and effective natural products that promote weight loss [128]. Several medicinal plants and their active compounds have shown marked activity in the management of obesity and its related diseases and, therefore, can be considered as a potential source for future natural drugs. In a recently published systematic review and meta-analysis of clinical trials aimed to evaluate the efficacy, safety, and mechanisms of effective herbal medicines in the management and treatment of obesity and metabolic syndrome in humans. Revealed that a total of 279 relevant clinical trials were published by the end of May 2019. Herbals containing green tea, *Phaseolus vulgaris*, *Zingiber officinale*, *Garcinia cambogia*, *Nigella sativa*, *Punica granatum*, olive oil, puerh tea, *Irvingia gabonensis*, Turmeric, and *Caralluma fimbriata* and their active ingredients were found to be effective in the management of obesity and related diseases [129].

The detection and purification of specific immunomodulatory plant compounds has the potential to reduce the side effects and high cost of conventional drugs. The objective of this current comprehensive review is to critically evaluate the potential of the most commonly used medicinal plants in the treatment of obesity-related inflammation based on clinical trials. It is important to consider the limitations and challenges that arise in the use of immunomodulatory medicinal plants and their isolated secondary plant compounds. The main limitations in the use of these natural products include: (1) Since the nature and concentrations of secondary plant metabolites are largely dependent on environmental conditions, the reproducibility of results obtained with plant extracts is inconsistent. This can be reduced if the standardization of extracts and their purification are thoroughly practiced. Although it is well documented that microbial endotoxin can alter the immune response, most anti-inflammatory studies conducted to measure the efficacy of medicinal plants have not adequately controlled bacterial contamination. (2) Since the concentration of phytochemicals and natural products is insufficient for development and clinical use,

phytochemists need to pay more attention to the development of innovative purification methods to increase their content for pharmaceutical applications. (3) The low availability of phytochemicals may explain their good toxicity profile, therefore, the improvement of delivery systems must be carried out cautiously. Despite significant advances in the pharmacological benefits of many natural products, diets, medicinal plants, and their active ingredients, they are still far from clinical use.

**Funding:** This research received no external funding.

**Institutional Review Board Statement:** Not applicable.

**Informed Consent Statement:** Not applicable.

**Data Availability Statement:** Not applicable.

**Conflicts of Interest:** The author confirms that this chapter's content has no conflict of interest.

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
