# Peer review of "Prevention and Treatment of Obesity-Related Inflammatory Diseases by Edible and Medicinal Plants and Their Active Compounds"

_2673-5601, doi:10.3390/immuno2040038_

Round 1

Reviewer 1 Report

The manuscript entitled, ‘Prevention and treatment of obesity-related inflammatory diseases by Greco-Arab and Islamic medicine-based diets and herbs’ highlights the role of Greco-Arab diets and medicinal plants in the management of inflammation associated with obesity. However, some major concerns should be addressed by the author prior to any possible consideration of this manuscript to be published in the journal of ‘immuno’.

1.      It would be better if the author change the title of the manuscript. The title sounds like research article and also the diet and eating habits are majorly associated with the geographical location rather the religious identity. (By Islamic medicine-based diet, does the author mean medicines reported in Islamic literature?)

2.      Writing font (eg., Times New Roman) should be consistent throughout the manuscript.

3.      The abbreviated words should be explained at their first use in the manuscript and then should be consistent throughout the manuscript.

4.      As author has mentioned in the Abstract “the objective of this comprehensive review is to critically evaluate the potential of the most used herbs in the management of obesity-related inflammation based on clinical trials.” The author should prepare a table compiling the herbal compounds that are in clinical trials to better represent the research progress on this field.

5.      The author could include a pictorial representation of pathways involved in the obesity related diseases and its management by targeting these pathways using the herbal compounds.

6.      The author must clarify the limitation of this study and the perspective regarding the future application of these medicinal compounds in controlling obesity-related inflammatory diseases.

7.      The section of ‘Conclusion’ is very vague. The ‘Conclusion’ section should be rewritten highlighting the important findings and novelty of this work.

Author Response

We thank the reviewer of his constructive comments. We changed the manuscript accordingly. 

The manuscript entitled, ‘Prevention and treatment of obesity-related inflammatory diseases by Greco-Arab and Islamic medicine-based diets and herbs’ highlights the role of Greco-Arab diets and medicinal plants in the management of inflammation associated with obesity. However, some major concerns should be addressed by the author prior to any possible consideration of this manuscript to be published in the journal of ‘immuno’.

  1. It would be better if the author change the title of the manuscript. The title sounds like research article and also the diet and eating habits are majorly associated with the geographical location rather the religious identity. (By Islamic medicine-based diet, does the author mean medicines reported in Islamic literature?)

Title changed to “ Prevention and treatment of obesity-related inflammatory diseases by edible and medicinal plants and their active compounds”

  1. Writing font (eg., Times New Roman) should be consistent throughout the manuscript.

Done

  1. The abbreviated words should be explained at their first use in the manuscript and then should be consistent throughout the manuscript.

Done

  1. As author has mentioned in the Abstract “the objective of this comprehensive review is to critically evaluate the potential of the most used herbs in the management of obesity-related inflammation based on clinical trials.” The author should prepare a table compiling the herbal compounds that are in clinical trials to better represent the research progress on this field.

I agree with the reviewer. However, in a recently published systematic review and meta-analysis of clinical trials aimed to evaluate the efficacy, safety, and mechanisms of effective herbal medicines in the management and treatment of obesity and metabolic syndrome in human revealed that a total of 279 relevant clinical trials were published by the end of May 2019. Therefore, I think that such a very large Table will confuse the readers.

We added this information in the concluding remarks

  1. The author could include a pictorial representation of pathways involved in the obesity related diseases and its management by targeting these pathways using the herbal compounds.

The requested info is included now in Figure 1

  1. The author must clarify the limitation of this study and the perspective regarding the future application of these medicinal compounds in controlling obesity-related inflammatory diseases.

Added in the concluding remarks

  1. The section of ‘Conclusion’ is very vague. The ‘Conclusion’ section should be rewritten highlighting the important findings and novelty of this work.

The concluding remarks have been written according to the reviewers comments

Reviewer 2 Report

The article is focused on a very interesting and popular subject. Many articles are available in the same area, so the objective of a new and original review is challenging!!

I will comment during my reading and suggest some modifications who may increase the final level of this exiting review and questions

line 96 mistake figure 1 adipnectins

line 101 remove one MCP1 from the list

Line 188: you explain that curcumin have a low toxicity please add a comment about the fact that this low toxicity may be due to the low bioavailability therefore increase of biodisponibility must be done cautiously. (See the conclusion part)

Paragraphs 3.1 and 3.2 have the same title please modify this point.

Line 190: I don’t understand the sentence « Curcuma longa rhizomes (Turmeric) are the source of white, green, and black tea. »?

Line 193 typing mistake: « than10 years »

Line 199: please add a comment about the fact that the effects are low so statistical confirmation is especially difficult in rather healthy people (see conclusion part)

Line 233 MCP1 was defined before so avoid repeat

Line 251 it is not clear for me that 6-shogaol is an inhibitor of PPAR gamma is it an activator? see https://doi.org/10.18632%2Foncotarget.16719 please comment this point.

Figure 2 typing mistake on thymoquinone.

Line 292: please add a reference to illustrate this sentence

Line 331: I don’t understand the sentence « which in turn increases the response to acetylcholine in a resistance artery, »

Line 357 please define « MetS »

Conclusion part: this is a major part of the article… I think that you will increase the level of your work with some complementary comments.

I suggest

a comment on the Caloric intake aspect as this point is largely related with obesity, longevity so how do you integrate the influence of caloric intake?

a comment on the difficulties of clinical development as many studies were done in rather healthy humans, so it is difficult to obtain a significant pharmacology activity. This is because natural compounds are not drugs…

a comment about the fact that the activities of extracts may be due to mixtures of structures so purification may induce a decrease of final activities…

Line 548 : please be more cautious as low availability may explain good toxicity profile…

Ref 96 please add the date

Complementary Question:

Explain in the introduction part the reasons for the selection of Figures on some natural compound and not the others?

Is it possible to organize the Figures according to a common global structure?

Good job!!

Author Response

I thank the reviewer for his helpful comments and suggestion. The manuscript has been edited to adapt these suggestions

The article is focused on a very interesting and popular subject. Many articles are available in the same area, so the objective of a new and original review is challenging!!

I will comment during my reading and suggest some modifications who may increase the final level of this exiting review and questions

line 96 mistake figure 1 adipnectins

Corrected

line 101 remove one MCP1 from the list

Removed

Line 188: you explain that curcumin have a low toxicity please add a comment about the fact that this low toxicity may be due to the low bioavailability therefore increase of biodisponibility must be done cautiously. (See the conclusion part)

Done

Paragraphs 3.1 and 3.2 have the same title please modify this point.

Done

Line 190: I don’t understand the sentence « Curcuma longa rhizomes (Turmeric) are the source of white, green, and black tea. »?

Corrected

Line 193 typing mistake: « than10 years »

Corrected

Line 199: please add a comment about the fact that the effects are low so statistical confirmation is especially difficult in rather healthy people (see conclusion part)

Done

Line 233 MCP1 was defined before so avoid repeat

Done

Line 251 it is not clear for me that 6-shogaol is an inhibitor of PPAR gamma is it an activator? see https://doi.org/10.18632%2Foncotarget.16719 please comment this point.

Corrected

Figure 2 typing mistake on thymoquinone.

Corrected

Line 292: please add a reference to illustrate this sentence

Done

Line 331: I don’t understand the sentence « which in turn increases the response to acetylcholine in a resistance artery, »

Corrected

Line 357 please define « MetS »

Corrected

Conclusion part: this is a major part of the article… I think that you will increase the level of your work with some complementary comments.

I suggest

The concluding remarks have been written according to the reviewer's comments

a comment on the Caloric intake aspect as this point is largely related with obesity, longevity so how do you integrate the influence of caloric intake?

a comment on the difficulties of clinical development as many studies were done in rather healthy humans, so it is difficult to obtain a significant pharmacology activity. This is because natural compounds are not drugs…

a comment about the fact that the activities of extracts may be due to mixtures of structures so purification may induce a decrease of final activities…

Line 548 : please be more cautious as low availability may explain good toxicity profile…

Sentence rewritten

Ref 96 please add the date

Added

Complementary Question:

Explain in the introduction part the reasons for the selection of Figures on some natural compound and not the others?

Added

Is it possible to organize the Figures according to a common global structure?

Done

Good job!!

Round 2

Reviewer 1 Report

Accept

Reviewer 2 Report

Thank you for the modifications,

Line 549 : mistake in the sentence : « syndrome in human. Revealed that a total of 279 »…

Please modify the structure of references for some of them year of publication is at the beginning (example 1 to 30) and then the year of publication is at the end …